# Fine-Tuning with Divergent Chains of Thought Boosts Reasoning Through Self-Correction in Language Models

## Abstract

Requiring a large language model to generate intermediary reasoning steps has been shown to be an effective way of boosting performance. In fact, it has been found that instruction tuning on these intermediary reasoning steps improves model performance. In this work, we present a novel method of further improving performance by requiring models to compare multiple reasoning chains before generating a solution in a single inference step. We call this method *Divergent CoT* (DCoT). We find that instruction tuning on DCoT datasets boosts the performance of even smaller, and therefore more accessible, LLMs. Through a rigorous set of experiments spanning a wide range of tasks that require various reasoning types, we show that fine-tuning on DCoT consistently improves performance over the CoT baseline across model families and scales (1.3B to 70B). Through a combination of empirical and manual evaluation, we additionally show that these performance gains stem from models generating multiple *divergent* reasoning chains in a single inference step, indicative of the enabling of *self-correction* in language models. Our code and data are publicly available.[1]

## 1 Introduction and Motivation

Chain of Thought (CoT; Wei et al. 2022), the prompting method to generate intermediate reasoning steps to answer a question, is recognized as a simple yet effective mechanism for improving the performance of large language models (LLMs). Given that requiring models to generate intermediary steps improves performance, it stands to reason that requiring models to simultaneously generate multiple chains could further improve performance. Prior work exploring this idea includes that by Wang et al. (2023), wherein they generate multiple CoTs and ensemble them with a voting mechanism. However, this and similar extensions (also see Section 2) do not use multiple inference chains *simultaniously*, and so the models do not have access to the different possible reasoning chains in a single inference step.

We present a novel mechanism that allows an LLM to compare multiple reasoning chains in *a single inference step*, leading to improved performance. We call this method Divergent Chain of Thought (DCoT). This method is inspired by the psychological theory of *Divergent and Convergent Thinking*, which posits that problem solving involves two distinct phases: divergent thinking, where many ideas are generated and explored, followed by convergent thinking, which involves considering these different ideas to arrive at a single solution or response Guilford (1967).

Unfortunately, the added complexity of generating multiple chains of thought (divergence) before selecting a single solution (convergence) makes this process too complex for most LLMs to perform using prompting alone. Our experiments show that the errors that are a result of the added complexity of this method almost completely offset the gains it might provide even in the most powerful current generation models, including GPT-4o. However, given that instruction fine-tuning, which involves fine-tuning on datasets consisting of task requirements and associated solutions, improves performance on those tasks, we hypothesize that similar instruction tuning on this complex divergent CoT is likely to enable not only large models but also smaller models to perform better. This

---

[1]https://anonymous.4open.science/r/DCoT-149B/

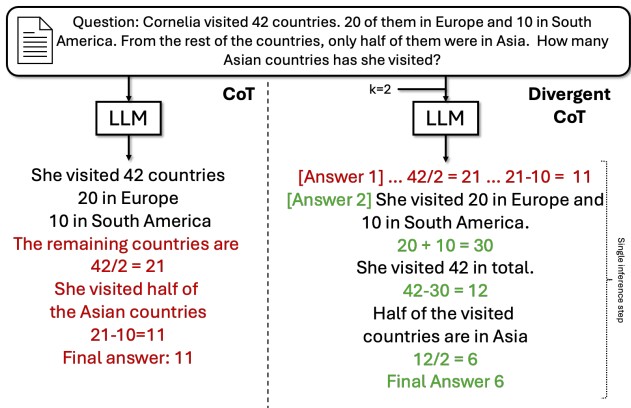

Figure 1: Divergent CoT ($k = 2$) generates $k$ CoTs in a single inference step and selects the correct answer.

hypothesis is further supported by previous results showing that the addition of CoTs into the instruction tuning data allows the model to better learn to use CoTs in generating outputs (Chung et al., 2024; Kim et al., 2023). As such, this work focuses on boosting the performance of LLMs, including small-scale, more easily accessible LLMs, by inducing them to generate accurate and effective DCoTs through instruction fine-tuning.

We demonstrate that fine-tuning using DCoTs improves LLM performance over the CoT baseline by rigorously testing on a range of tasks requiring different types of reasoning across model families and scales (1.3B to 70B). Moreover, we show that DCoT fine-tuning provides the additional benefit of allowing LLMs to improve their first answer without external feedback, which we verify through a manual analysis of the outputs. Additionally, we show that once fine-tuned, DCoT can be further augmented by the same methods that boost CoT, such as self-ensembling (Wei et al., 2022). Independently, performance boosts provided by instruction tuning on DCoT data show that we can encode other non-trivial reasoning methods into LLMs by instruction tuning on appropriate datasets.

The contributions of this work are as follows:

- We introduce *Divergent CoT*, a modification to CoT that generates multiple reasoning chains and selects an answer in a single inference step.
- We show the effectiveness of fine-tuning on DCoT data, through a rigorous set of experiments on a range of LLM families and sizes across multiple multiple reasoning tasks.
- We show DCoT has the side-effect of learning to *self-correct* without external feedback or prompt optimization, which to the best of our knowledge, is the first work to do so.

## 2 RELATED WORKS

In this section, we examine related work from three distinct perspectives: (i) prompting methods that enhance CoT prompting for divergence, (ii) research focused on instruction tuning models using CoTs, and (iii) studies on self-correction.

**Divergent Prompting.** Many works have shown the benefits of generating diverse CoTs and aggregating them (Wang et al., 2023; Zhang et al., 2024; Yoran et al., 2023; Li et al., 2022; Weng et al., 2023; Zhao et al., 2023a;b). In particular, Wang et al. (2023) proposed the creation of *self-assembles* of CoTs to improve LLM's performance, which they call self-consistency. They sample a series of CoTs, select the most repeated answer, and show large performance gains on reasoning tasks. Yoran et al. (2023) extends this work by creating a meta prompt that aggregates the reasoning paths instead of selecting the most common answer. Zhang et al. (2024) propose explicit steps to contrast each CoT and reflect on the final answer. However, none of these works induce LLMs to generate multiple CoTs in the same inference step.

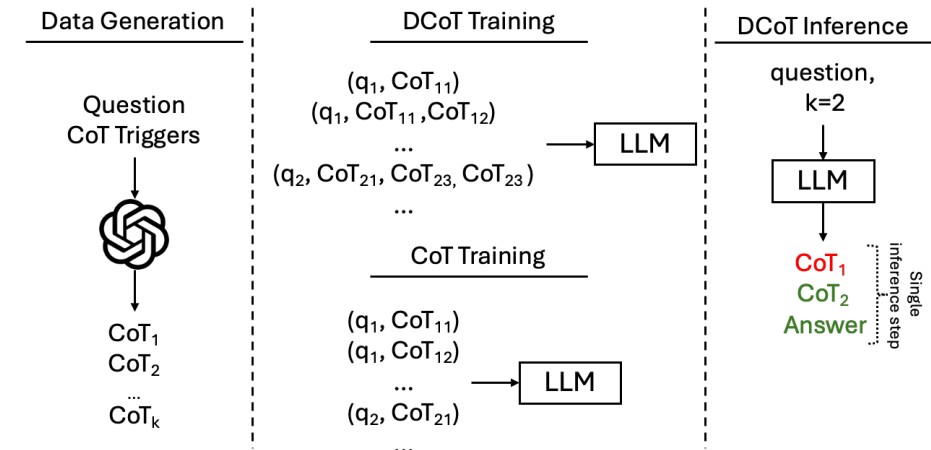

Figure 2: We train on a series of CoTs to make the model learn how to generate multiple CoTs in one inference step.

**Divergent Fine-Tuning.** The success of CoT prompting led to the creation of instruction-tuning datasets with CoTs (Chung et al., 2024). Kim et al. (2023) argue that small LMs perform poorly on CoT on unseen tasks compared to large LMs. Hence, they create an instruction-tuning dataset of CoT to equip small LMs with CoT capabilities. Others suggest distilling CoTs from very large language models (vLLMs) (Hsieh et al., 2023; Li et al., 2023a). Ho et al. (2023) also show the benefits of distilling CoTs from these vLLMs and claim that sampling multiple CoTs per question is an effective data augmentation technique that improves the performance of distilled models. However, they do not use this diversity at inference time, and unlike us, their method only generates one CoT per question. Huang et al. (2023) show that vLLMs can improve performance on reasoning tasks by self-training on their own CoT generations from sampling.

**Self-Correction.** Huang et al. (2023) defines it as the ability of an LLM to correct its initial response without relying on external feedback. Some initial works suggest that LLMs possess self-correct abilities (Shinn et al., 2024; Madaan et al., 2023; Pan et al., 2023; Kim et al., 2024; Weng et al., 2023; Jiang et al., 2023). However, Huang et al. (2024); Stechly et al. (2024); Tyen et al. (2023) argue that self-correction's gains stem from the use of external feedback. *Divergent CoT*, on the other hand, exhibits superior performance when generating more than one CoT in a single inference step, using essentially the same prompt, suggesting that DCoT may enable models to self-correct without external supervision or prompt optimization.

## 3 METHODS

To analyze the effectiveness of DCoT, we first evaluate the performance of LLMs when prompted to generate multiple chains. However, we focus the majority of our experiments on the effect of instruction tuning on DCoTs, as this allows us to extend the effectiveness of our methods to smaller, more accessible models.

### 3.1 PROMPTING

We conducted exploratory experiments to evaluate the effectiveness of DCoT prompting on commercial black-box LLMs. We use prompts to require models to generate multiple CoTs, compare them, and generate an answer, all in a single inference step. We found that smaller LLMs, with fewer than 100B parameters, lacked the capacity to perform this complex task. When prompted, they often generated the same CoT repeatedly. Even when they did generate multiple CoTs, our manual evaluation revealed they failed to effectively select the correct answer from among them. These results are in line with prior results that indicate that these smaller models are also not the most effective in generating CoTs (Kim et al., 2023). While GPT-4o showed more success, the complexity of the

task also heightened its tendency to hallucinate. Consequently, we observed no performance boost through prompting alone and thus focused our experiments on instruction tuning using DCoTs, as detailed in subsequent sections. Appendix C reports the prompts we used.

## 3.2 FINE-TUNING

**DCoT.** We aim to instruction-tune LLMs to generate a sequence of divergent CoTs before selecting the final answer in a single inference step at inference time. To this end, we devise a DCoT instruction template, where we introduce a set of commands (in brackets) to request the number of CoTs to generate:

**Prompt**: [Question] Question [Options] Options [Number of answers] $k$
**Response**: [Answer 1] $CoT_1$ [Answer 2] ... [Answer k] $CoT_k$ [Final answer] answer

We instruction-tune each of the models we experiment with (Section 3.5) using the above template. We generate DCoT data in the required format using methods described in Section 3.3. For brevity, we refer to instruction-tuned models on DCoT data as `DCoT`.

**CoT (Baseline).** So as to establish a comparable baseline, we instruction-tune the same LLMs using the more traditional CoT format. To ensure a fair comparison, we use the same reasoning chains as above. As shown in Figure 2, each data point is composed of a question and a CoT, and a question may appear in more than one data point but with a different CoT. In this way, the model leverages CoT diversity at training time but, unlike in `DCoT`, it does not do so at inference time. Once again, for brevity, we refer to these models as `CoT`.

## 3.3 DATASET GENERATION

We follow the methods set out by Ott et al. (2023) to create CoTs that we use to create our CoT and DCoT tuning datasets. We use GPT 3.5 turbo in the zero-shot setting with multiple triggers to generate CoTs. Specifically, *CoT Triggers* are prompt suffixes, such as *"Let's think step by step"* that 'trigger' LLMs to generate CoTs. We use the same triggers as in (Ott et al., 2023). For each question, we select four random CoT triggers. We limit the number of CoTs to four to ensure that the targets fit the context window of the LLMs. We restrict the training data to those reasoning chains that lead to correct answers as determined by the labels provided by the corresponding dataset. We report the prompt templates and triggers in Appendix H.

## 3.4 FINE-TUNING DATASET CREATION

Table 1 lists the datasets we use to generate our CoTs and train the models. These datasets were selected following prior works (Wang et al., 2023; Yoran et al., 2023). We have added BoardgameQA (Kazemi et al., 2023) to include logic and ConditionalQA (Sun et al., 2022) to include natural conditional reasoning, both of which are highly complex and a *second thought* can be beneficial to find the answer. With this selection, we cover multiple domains, output spaces, and reasoning abilities. More details are provided in Appendix A.

Table 1: Brief description of the training datasets.

| Dataset | Reasoning Type |
|---|---|
| ARC (Clark et al., 2018) | High-School Science |
| BGQA (Kazemi et al., 2023) | Logic |
| CoinFlip Wei et al. (2022) | State-tracking |
| CondQA (CQA; Sun et al. 2022) | Conditional |
| GSM8K (Cobbe et al., 2021) | Math |
| HotpotQA (HQA; Yang et al. 2018) | Explicit multli-hop |
| LLC (Wei et al., 2022) | Symbolic |
| Quartz (Tafjord et al., 2019) | Relationships |
| StrategyQA (StrQA; Geva et al. 2021) | Implicit multi-hop |

## 3.5 MODELS

We train a series of models covering the scaling laws and different families. Concretely, we employ Phi 1.5 (1.3B; Li et al. 2023b), Phi 2 (2.7B; Abdin et al. 2023), LLaMA-2 7B, LLaMA-2 13B (Touvron et al., 2023). For all of our experiments, we select the non-instruction tuned-based models so as to ensure that the comparison between DCoT and CoT is fair. This is because instruction-tuning datasets contain CoT data (Touvron et al., 2023), which would otherwise make the comparison unfair. We also conduct a smaller experiment on LLaMA-2 13B Chat to analyze whether our DCoT instruction-tuning method can be applied to already-instruction-tuned models and on LLaMA-2 70B. We refer the reader to Appendix B for details on the training setup of the models.

## 3.6 EVALUATION

We use the macro average F1 metric for all in-domain classification tasks and the squad-metric (Rajpurkar et al., 2016) for the in-domain span-extraction tasks (i.e., ConditionalQA and HotpotQA). We run our DCoT with $k \in [1, 4]$ and select the best $k$ for each dataset based on the dev set. For LLaMA-2 70B, we only report results on the dev set due to the costs for hyperparameter tuning. Further discussions are provided in Appendix B.

For the out-of-domain evaluation, we select tasks from the three domains on which self-consistency has been shown to improve, namely math, commonsense, and symbolic reasoning (Wang et al., 2023). Specifically, we evaluate on AQuA (math; Ling et al. 2017), SVAMP (math; Patel et al. (2021)), CommonsenseQA (CSQA; Talmor et al. 2019), and Object Counting (symbolic reasoning; Suzgun et al. 2023). We hypothesize that DCoT tuning will improve performance on these tasks.

Lastly, we use Big Bench Hard (Suzgun et al., 2023) as a control experiment to evaluate whether generating multiple CoTs can confuse the models and generate worse performance. We specifically use this benchmark because their authors report that CoT is only beneficial in large enough models; in other words, not using CoT is better for small models. This implies that it is extremely difficult for small models to generate correct CoTs for these tasks, and therefore, generating more than one is even more difficult, so it is reasonable to question whether DCoT can reduce performance.

## 4 RESULTS AND ANALYSIS

In this section, we present results demonstrating the following:

1. The in-domain effectiveness of DCoT, as measured by its effectiveness on the tasks that we instruction tune on (Section 4.1)
2. The generalizability of DCoT to unseen tasks (Section 4.2)
3. The robustness of DCoT to tasks where CoT is detrimental (Section 4.3)
4. The feasibility of using post-hoc CoT extensions with DCoT (Section 4.4)
5. That DCoT elicits *self-correct* abilities in LLMs (Section 5 and 5.2)

## 4.1 DCoT IS BENEFICIAL ON IN-DOMAIN TASKS

**Overall performance.** The first two rows of each model in Table 2 compares DCoT with the CoT baseline using the greedy decoding.[2] As explained in Section 3.6, DCoT uses the best $k$ for each dataset according to the results on the dev set. The first result we observe is that DCoT achieves consistent and significant performance gains compared to CoT. The largest average gain is 2.19 for Phi 1.5, the smallest gain is 1.75 for Phi 2, and the maximum gain is 7.59 for Phi 2 on BGQA. We also observe that, overall, these gains are consistent across all datasets for all models. In particular, we only observe one dataset where CoT outperforms DCoT in Phi 1.5 and Phi 2, two in LLaMA 7B, and three in LLaMA-2 13B. However, the largest decrements are on StrategyQA, the only boolean QA dataset. We attribute this to the nature of this dataset, where only two options are possible, and thus, the divergence in the reasoning is less needed.

---

[2]CoinFlip results are omitted because all models achieve perfect scores.

Table 2: Comparison of `DCoT` against `CoT` on the test sets. *70B results on the dev set.

| LLM | Method | Avg. | ARC | BGQA | CQA | GSM8K | HQA | LLC | Quartz | StrQA |
|---|---|---|---|---|---|---|---|---|---|---|
| Phi 1.5 | CoT | 47.20 | 48.70 | 32.39 | 61.21 | 34.95 | 32.56 | **41.00** | 72.69 | 54.08 |
| | DCoT (Ours) | **49.39** | **50.01** | **38.60** | **62.48** | **36.85** | **34.81** | 39.00 | **77.39** | **55.97** |
| (1.3B) | CoT + SC | 46.48 | **53.81** | 21.59 | 63.39 | **40.33** | 33.63 | 32.00 | 75.11 | 51.96 |
| | DCoT + SC | **49.01** | 53.24 | **27.60** | **65.23** | 40.18 | **37.79** | 31.00 | **81.08** | **55.97** |
| Phi 2 | CoT | 60.85 | 70.87 | 39.48 | 65.13 | 56.71 | 52.65 | 58.00 | 82.91 | **61.06** |
| | DCoT | **62.60** | **73.77** | **47.07** | **68.61** | **60.73** | **55.15** | 58.00 | **83.16** | 54.34 |
| (2.7B) | CoT + SC | 61.50 | 74.36 | 28.99 | 68.14 | 64.97 | 55.82 | 55.00 | 85.20 | **59.51** |
| | DCoT + SC | **65.12** | **76.06** | **44.16** | **70.53** | **68.08** | **58.61** | **66.00** | **86.09** | 51.43 |
| LLaMA2 | CoT | 58.97 | 61.63 | **43.13** | 65.73 | 28.51 | 53.88 | 75.00 | 79.32 | **64.59** |
| | DCoT | **60.80** | **62.70** | 41.91 | **70.99** | **29.57** | **56.26** | **82.00** | **81.37** | 61.64 |
| 7B | CoT + SC | **62.90** | 65.98 | **46.04** | 69.92 | 33.97 | 57.05 | 81.00 | 83.28 | **65.99** |
| | DCoT + SC | 61.09 | **68.53** | 28.20 | **71.36** | **36.01** | **58.35** | **83.00** | **84.05** | 59.22 |
| LLaMA2 | CoT | 64.39 | **71.79** | 42.63 | 70.25 | 42.53 | 60.23 | **81.00** | **84.82** | 61.85 |
| | DCoT | **66.18** | 71.41 | **50.21** | **71.56** | **44.28** | **63.52** | 80.00 | 83.29 | **65.16** |
| 13B | CoT + SC | 66.82 | 74.82 | 40.80 | **72.72** | 50.27 | 62.34 | 80.00 | **85.84** | 67.75 |
| | DCoT + SC | **68.12** | **74.89** | **41.27** | 72.61 | **54.51** | **65.92** | **86.00** | 85.07 | 64.65 |
| LLaMA2 | CoT | **64.87** | 70.43 | **44.39** | **71.71** | 42.76 | 60.83 | **78.00** | 84.04 | 66.78 |
| 13B Chat | DCoT | 64.62 | **72.22** | 40.94 | 71.59 | **44.20** | **63.87** | 71.00 | **85.43** | **67.68** |
| LLaMA2 | CoT | 66.96 | 81.69 | **44.34** | **73.59** | 56.00 | **55.94** | 76.00 | 81.99 | 66.15 |
| 70B* | DCoT | **68.63** | **89.04** | 38.30 | 69.57 | **66.00** | 49.78 | **82.00** | **85.99** | **68.34** |

Table 3: `DCoT` average performance across different number of CoTs per question on the dev sets.

| LLM | k=1 | k=2 | k=3 | k=4 |
|---|---|---|---|---|
| Phi 1.5 | 49.64 | 49.36 | 49.16 | 48.47 |
| Phi 2 | 61.60 | 63.04 | 64.21 | 62.71 |
| LLaMa2 7B | 61.08 | 62.20 | 62.28 | 62.26 |
| LLaMA2 13B | 65.37 | 67.85 | 67.45 | 67.32 |

**Performance across $k$.** Table 3 shows the average performance across all datasets for each $k$. Here, we focus on efficiency and aim to obtain gains with small $k$. We can see that, in general, a $k > 1$ (i.e., the number of generated CoTs in our `DCoT`) improves the performance of the model across all datasets (compared to $k = 1$). This also shows that we do not need to optimize $k$ to make DCoT effective, any $k > 1$ provide performance gains. Furthermore, we observe gains even with $k = 2$, showing the efficiency of our approach. The best performance of our model is achieved with more than one CoT in 25 cases out of 32 dataset $\times$ LLM combinations (see Figure 3 in Appendix G). However, `DCoT` sometimes exhibits some performance drop when increasing $k$ (e.g., Phi-2@4 on GSM8K). We attribute this to an *overthinking* effect, where the model tries to explore more CoTs and ends up generating wrong CoTs that bias the final answer. We report the best $k$ for each dataset $\times$ LLM combination on Table 15 in Appendix F.

**`DCoT@1` $\approx$ `CoT`** Table 12 in Appendix D reports the mean and standard deviation of both methods across three random seeds on the dev set. An important phenomenon we observe there is that the performance of `DCoT` when generating a single CoT (i.e., `DCoT@1`) is very similar to the `CoT` baseline, as expected. This result shows that our `DCoT` training does not interfere with the regular CoT generation. *Therefore, `DCoT` is a safe replacement to `CoT` in regular instruction-tuning datasets.*

We also conduct a smaller experiment on general instruction-tuned models (LLaMA2 13B chat). It is worth noting that comparing `CoT` with `DCoT` is not completely fair in this setting because this model has already been fine-tuned on CoTs (Touvron et al., 2023); thus, the `CoT` training is

Table 4: `DCoT` vs. `CoT` on unseen tasks.

| LLM | Method | AQuA | CSQA | ObjCnt | SVAMP |
|---|---|---|---|---|---|
| | CoT | 20.27 | 33.88 | **35.60** | 40.00 |
| Phi 1.5 | DCoT@1 | 21.51 | 32.26 | 25.20 | **40.50** |
| | DCoT@2 | 17.31 | 34.23 | 27.60 | 30.00 |
| | DCoT@3 | **22.38** | 33.81 | 30.80 | 30.00 |
| | DCoT@4 | 22.06 | **34.73** | 30.00 | 31.50 |
| | CoT | 29.52 | 44.29 | 54.00 | 55.00 |
| Phi 2 | DCoT@1 | **34.86** | 44.15 | **58.40** | **60.50** |
| | DCoT@2 | 34.09 | 44.13 | 56.40 | 60.50 |
| | DCoT@3 | 31.83 | **45.99** | 57.60 | 60.00 |
| | DCoT@4 | 34.73 | 45.43 | 56.40 | 59.50 |
| | CoT | **19.41** | 38.41 | 34.80 | 39.50 |
| LLaMA2 7B | DCoT@1 | 17.70 | 36.94 | **40.00** | 41.50 |
| | DCoT@2 | 17.27 | **40.79** | 39.60 | **43.00** |
| | DCoT@3 | 16.90 | 40.67 | 36.80 | 43.00 |
| | DCoT@4 | 17.21 | 40.43 | 37.20 | 39.00 |
| | CoT | **24.85** | 46.55 | 45.2 | **62.50** |
| LLaMA2 13B | DCoT@1 | 23.98 | 44.62 | 46.00 | 55.00 |
| | DCoT@2 | 22.42 | 45.48 | 47.60 | 53.50 |
| | DCoT@3 | 20.72 | **47.42** | 52.40 | 56.50 |
| | DCoT@4 | 23.13 | 46.45 | **54.00** | 53.50 |

Table 5: Results on Big Bench Hard.

| Method | Phi 1.5 | Phi 2 | LLaMA2 7B | LLaMA2 13B |
|---|---|---|---|---|
| CoT | 28.37 | 46.7 | 31.08 | 36.38 |
| DCoT@1 | 28.31 | 44.56 | 31.23 | 34.59 |
| DCoT@2 | 28.07 | 45.81 | 31.11 | 35.94 |
| DCoT@3 | 28.35 | 45.92 | 31.00 | 36.90 |
| DCoT@4 | 28.21 | 46.71 | 31.13 | 36.45 |

larger and more diverse than the `DCoT` one. Despite this, we observe that in more than half of the datasets `DCoT` outperforming `CoT`. However, the average score across all tasks is very similar for both methods. This is because of the performance outlier in LLC, where `CoT` outperforms `DCoT` by 7 points.

## 4.2 DCoT is Beneficial on Unseen Tasks

In this section, we investigate whether `DCoT` remains beneficial on unseen tasks. To answer this, we utilize the `DCoT` and `CoT` trained on the nine tasks described on Section 3.4 and evaluate them on new ones where self-consistency is known to improve performance (Wang et al., 2023). We report these results in Table 4 and observe that `DCoT` outperforms `CoT` on most datasets with Phi 1.5, Phi 2, and LLaMA2 7B. In particular, we find gains larger than 5 points on AQuA and SVAMP for Phi 2, and larger than 3 on ObjCnt for Phi2 and SVAMP for LLaMA-2 7B. However, the results on LLaMA-2 13B are mixed and only on the non-math domains we observe significant gains. Moreover, we observe consistent and large gains by increasing $k$ on Object Count, showing its capability to improve the CoTs consistently.

### 4.3 DCoT is Robust on Tasks where CoT is Detrimental

We analyze the performance of our method on Big Bench Hard, a benchmark where small models do not benefit from CoTs (Suzgun et al., 2023) to discover whether generating multiple CoTs can further confuse the models and lead to worse results than the CoT baseline. The results from Table 5 show that on these tasks, DCoT exhibits similar performance to CoT, thus demonstrating that DCoT does not lead to deterioration in challenging cases, where CoT might be detrimental. Moreover, we can observe some performance gains on Phi 2 and LLaMA-2 13B when increasing $k$, further showing the robustness of DCoT tuning and generalization to unseen tasks.

### 4.4 DCoT Benefits from CoT Extensions

The last two rows of each model (i.e., CoT+SC and DCoT+SC) in Table 2 compares our DCoT with the CoT baseline using the self-consistency decoding (Wang et al., 2023). This decoding method is an add-on that has been shown to increase the performance of CoT across a wide range of tasks by sampling multiple generations and the aggregating them by a voting mechanism.

We observe that our DCoT also benefits from this mechanism and keeps its performance gains over the CoT baseline, showing that our method can be a replacement for CoT in future instruction-tuning datasets. It is also worth noting that our DCoT with the greedy decoding even outperforms CoT+SC on all models, showing its superiority against CoT.

## 5 DCoT Elicits Self-Correct Abilities

*Self-correction* is the ability of an LLM to correct its initial response without relying on external feedback (Huang et al., 2023). According to them, truly intrinsic self-correction is yet to be found in LLMs. Our findings show that DCoT-tuned models, trained to generate a multiple CoTs where subsequent CoTs can provide different answers, *do* have the capacity to self-correct initial CoT answers without external feedback, a result of particular note since they are not explicitly trained to do so. In this section, we provide a detailed empirical and careful manual analysis to quantify this effect and understand why it is a form of self-correction.

### 5.1 Quantitative Analysis of Self-Correction

In the previous sections, we have demonstrated that DCoT does indeed improve performance. However, these gains could be achieved in two distinct ways: it could be a result of *self-ensembling* as in the case of self-consistency, or alternatively, it could be a result of *self-correction*. To test which of these mechanisms leads to improvements, we compare the performance of DCoT when we generate two reasoning chains ($k = 2$) to that where we generate just one. Importantly, any performance improvement between these cases cannot be a result of self-consistency as two outputs are not sufficient to provide a majority vote, and at least three reasoning chains are needed. For this reason, we only compare DCoT@1 vs DCoT@2, and we do not compare with DCoT@3 or 4.

We can see in Table 6 that all models improve performance for most datasets when generating two CoTs instead of one. Specifically, in over 62% of cases (i.e., 25 out of 40 LLM × dataset). Furthermore, we can observe performance improvements greater than 0.5 for more than half of the datasets for Phi 1.5, Phi2, LLaMA2 13B, and 70B. This result is significant because it means

Table 6: Performance gain from generating two CoTs instead of one on the dev set.

| LLM | ARC | BGQA | CQA | GSM8K | HQA | LLC | Quartz | StrQA |
|---|---|---|---|---|---|---|---|---|
| Phi 1.5 | 1.26 ↑ | 2.10 ↑ | 0.10 | 3.00 ↑ | 0.83 ↑ | -14.00 ↓ | 3.38 ↑ | 1.11 ↑ |
| Phi 2 | -3.56 ↓ | -2.38 ↓ | 0.95 ↑ | 0.80 ↑ | 1.06 ↑ | 14.00 ↑ | 1.55 ↑ | -0.85 ↓ |
| LLaMA2 7B | 1.28 ↑ | -0.99 ↓ | -0.56 ↓ | 4.00 ↑ | -0.01 | 6.00 ↑ | -1.04 ↓ | 0.25 |
| LLaMA2 13B | 4.15 ↑ | 0.91 ↑ | -1.02 ↓ | 3.00 ↑ | 2.02 ↑ | 12.00 ↑ | 0.77 ↑ | -2.03 ↓ |
| LLaMA2 70B | 3.24 ↑ | 1.38 ↑ | 3.68 ↑ | 10.00 ↑ | 0 | 4.00 ↑ | -1.00 ↓ | -4.07 ↓ |

that the generation of a second CoT is beneficial. In other words, the second CoT *overrides* the first, leading to the observed performance gains. We observe a similar effect on the unseen tasks in Table 4, although the effect is less pronounced due to lower overall improvements on these out-of-domain tasks. Regardless, across models and tasks, we find that in 6/16 cases (i.e., models × tasks), DCoT@2 improves over DCoT@1, and in 8/16 DCoT@k for $k > 1$ improves over DCoT@1, with an additional two cases where the drop with increased $k$ is only marginal.

These results indicate that DCoT tuning enables models to self-correct. Notably, our training data includes only reasoning chains that lead to the correct answer, never incorrect ones. This suggests that the ability to self-correct can be enabled in LLMs without explicitly training for it.

## 5.2 DCoT@2: MANUAL EVALUATION

We conduct a manual evaluation to verify our conclusions based on the quantitative results. Specifically, we verify that DCoT achieves self-correction abilities by generating an improved second CoT. To this end, we select instances for every dataset where LLaMA 7B with DCoT@1 outputs an incorrect answer while DCoT@2 results in a correct answer. We then randomly sample five instances per dataset, resulting in a total of 33 samples. We note that the first reasoning chain of DCoT@2 might differ from that of DCoT@1 because they are different runs. We find this to be the case in nine instances. This implies that in most cases, the first CoT is the same in both cases. Of these instances where the first reasoning chain is shared, we observe that in 45% of the cases, the second CoT of DCoT@2 exhibits a different reasoning pattern from the first. Therefore, in 45% of the cases, a second, improved CoT, allows the model to generate a correct answer, when the first CoT results in an incorrect answer. In other words, we observe that the performance gains in DCoT@2 can be attributed to *self-correction*.

A more fine-grained analysis of these instances reveals that in one case, we observe that the second CoT is very similar to the first one but extracts more information from the context and uses it for the logical inference that allows it to reach the correct answer. In three cases, the second CoT fixes a conclusion from the first CoT. In the last three cases, the CoTs lead to two potential answers, and only the second CoT selects the correct one. Table 13 in Appendix E shows examples of these observations. Overall, our manual analysis confirms that the performance gains achieved through DCoT result from the model self-correcting its initial answer.

## 5.3 DCoT@3: QUANTITATIVE AND QUALITATIVE ANALYSIS OF WRONG CORRECTIONS

In this section, we analyze how DCoT works when we generate three CoTs to confirm that DCoT generates CoTs that correct prior ones instead of doing a self-ensemble of CoTs. Firstly, we observe in Table 7 that DCoT@3 only incorrectly revises the CoTs in very minor cases.[3] For example, for ARC, this only happens in 19 cases out of the 168 (11%) cases where DCoT@1 is correct.

We have manually inspected 15 random examples of those wrong corrections and discovered three patterns: i̲) The second and the third answers are the same and wrong (e.g., answers are A,B,B). This happens in six cases. ii̲) The first and second answers are the same, while the third one is a wrong revision (e.g., A,A,B̲). This was found eight times. iii̲) All answers are different. We only found one case, and the final answer selected by the model was the second one. The first and second patterns show that our model tends to select the last revision as the final answer, even if the majority of answers point otherwise. This illustrates that DCoT is not a mere ensemble of CoTs, and instead, is trying to self-correct.

Table 7: Instances where DCoT@1 gives the correct answer and DCoT@3 does a wrong correction.

|  | ARC | BGQA | GSM8K | Quartz | StrQA |
|---|---|---|---|---|---|
| # @1 Correct | 168 | 224 | 158 | 310 | 162 |
| # @3 Wrong | 19 | 44 | 35 | 17 | 34 |

---

[3]We count as wrong corrections those cases where DCoT@3 generates a first CoT that is the same as in DCoT@1, but its final answer is different from the final one of DCoT@1, and thus wrong.

## 6 DISCUSSION

It is important to note that both `DCoT` and `CoT` are trained on exactly the same amount of CoTs and questions. While the `CoT` baseline uses data points in the form of $[(q, cot_1), (q, cot_2), ...]$, `DCoT` uses data points in the form of $[(q, cot_1, cot_2, ...) , ...]$. In other words, a simple re-organization of the training CoTs into the form of multiple CoTs per label has a major impact on the model's performance, making our results more striking. Importantly, `DCoT@1` matches the performance of the `CoT` baseline, indicating that it is safe to augment existing instruction-tuning datasets with `DCoT` data, as it will not hinder model performance.

`DCoT` is different from ensembling methods like self-consistency, which also benefit from generating multiple candidate answers but do so across different inference steps using high-temperature values. `DCoT`, while it may resemble these ensemble methods, is fundamentally different. Our method generates reasoning chains that have access to previous ones and shows performance improvements even when generating just two CoT chains.

The most surprising aspect of our findings is that `DCoT` has the ability to self-correct. This ability presents itself despite us not explicitly training models to learn to correct themselves. The reasoning chains we use for training are all correct CoTs, and we fine-tune base models without prior instruction-following capabilities. We believe this self-correction is the reason why our model performs best with smaller $k$. After one or two attempts to self-correct, it is highly unlikely the model will be able to find a correct revision. We also find this in (Madaan et al., 2023) and (Kim et al., 2024). Where the former uses $k \leq 4$ and the latter finds the optimal $k$ at 3. We argue that these abilities stem from the model's attempt to generate subsequent correct CoTs. In other words, the model may generate a first wrong CoT without knowing it, but it generates a second CoT that is correct and, therefore, as a side-effect, corrects the first one.

More generally, we deduce that these abilities arise from the model's capacity to learn to generalize based on the divergent reasoning chains we train on. This supposition gains further credence from recent work suggesting that instruction tuning allows models to generalize their abilities to solve tasks, rather than leading to novel capabilities Lu et al. (2023). Regardless of the underlying mechanism—identification of which we leave to future work—we provide a novel method for enabling LLMs to self-correct. We posit that instruction tuning on other complex multi-step reasoning problems, as we have done with generating multiple divergent CoTs before converging on a final answer, will lead to encoding those complex capabilities into LLMs while also allowing them to generalize in powerful new ways.

## 7 CONCLUSIONS

This work presents Divergent Chain of Thought (`DCoT`), a new CoT method that aims to improve LLM's performance on reasoning tasks by generating multiple CoTs in a single inference step. We show through extensive quantitative experiments the effectiveness and scalability of our method across a wide range of reasoning tasks (in-domain and out-of-domain), model families, and sizes. Furthermore, we show its efficiency w.r.t $k$ by achieving gains even with $k = 2$. We also show that `DCoT` can be extended with any CoT extension, such as self-consistency, wherein it outperforms `CoT` similarly extended with self-consistency. Lastly, we show a beneficial side effect of our method: the subsequent generated CoTs can self-correct previous reasoning chains without any external feedback or prompt optimization. This is the first work that achieves such *self-correct* ability in LLMs. We show quantitatively the occurrence of this phenomenon with gains up to 14 points, and further explain it with a qualitative analysis showing that the second generated CoT provides a different reasoning chain compared to the first one and that this second CoT leads to a correct answer. We leave as future work extending our DCoT fine-tuning to other types of prompting such as code prompting (Puerto et al., 2024) or graph of thoughts (Besta et al., 2024).

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

# A  DATASETS

All the datasets used in this work are exclusively in English language. In particular, we use ARC (Clark et al., 2018), BGQA (Kazemi et al., 2023),CoinFlip Wei et al. (2022), ConditionalQA (CQA) (Sun et al., 2022), GSM8K (Cobbe et al., 2021), HotpotQA (HQA) (Yang et al., 2018), LLC Wei et al. (2022), Quartz (Tafjord et al., 2019), and StrategyQA (StrQA) (Geva et al., 2021) for training, while we use AQuA (Ling et al., 2017), CommonsenseQA (Talmor et al., 2019), Object Count (a task of Big Bench Hard Suzgun et al. 2023), SVAMP (Patel et al., 2021), and Big Bench Hard for out of domain evaluation. For BGQA, we use the partition main-3, the most difficult one requiring 3-hop reasoning skills.

Some of these datasets do not provide a validation set. In those cases, we randomly sample 500 instances from the training set and use them as validation set. Similarly, when a dataset does not provide a test set, we use the validation set as a test and create a validation set from the unused instances from the training set. When the training set is not larger than 1k, we divide the validation set into two. For Last Letter Concatenation (LLC), the training set is very small (350 instances), and the test set is also very small (150), so we pick 50 instances of the test set as validation and 100 as test. We release in our github repository the exact partitions we used.

Table 10 reports the licenses and sizes of the training, dev, and test sets of the datasets we used and Table 11 reports for the out of domain datasets. We use these datasets for research purposes only, fulfilling their intended use.

Due to the large size of LLaMA-2 70B and its computation costs, we trained it on a smaller sample data of 900 questions. Similarly, for inference, we pick a random sample of 100 questions per dataset.

# B  EXPERIMENTAL SETUP

We run all our experiments on a GPU cluster with an Nvidia A180. To run GPT models, we use the Azure OpenAI service and prompt them with the library Langchain.[4] We use Scikit-learn (Pedregosa et al., 2011) for the implementation of the evaluation metrics.

We train all models using LoRA (Hu et al., 2022) with the PEFT library (Mangrulkar et al., 2022) and use vLLM (Kwon et al., 2023) as the inference engine. For training, we load the models with fp8, while for inference, we load them with fp16. We train models for three epochs, save checkpoints for each epoch and select the best checkpoint based on the average results on the dev set.

Due to the challenge of running very large models, such as LLaMA-2 70B, to simplify the evaluation setup. We trained the model with 8-bit quantization and ran the evaluation on 4-bit. Instead of evaluating on the full dev sets, we had to evaluate on a random sample of 100 questions per dataset and only evaluate the last checkpoint. Therefore, we could not conduct hyperparameter tuning either. Because of these challenges, we cannot report results on the test set, and instead, we only report results on the dev set. It is important to emphasize again that we do not conduct any hyperparameter

---

[4] https://github.com/langchain-ai/langchain

Table 8: Training parameters

| Param. name | Value |
|---|---|
| lora_r | 64 |
| lora_alpha | 16 |
| lora_dropout | 0.1 |
| batch size | 4 |
| max_grad_norm | 0.3 |
| learning_rate | 2e-4 |
| weight_decay | 0.001 |
| optim | paged_adamw_32bit |
| lr_scheduler_type | constant |
| max_steps | -1 |
| warmup_ratio | 0.03 |
| group_by_length | True |
| max_seq_length | 4096 |
| packing | False |
| seeds | 0, 42, 2024 |
| load_in_8bit | True |

Table 9: Best hyperparameters tuned on the dev set.

| Model | Method | Seed | Epoch |
|---|---|---|---|
| Phi 1.5 | CoT | 0 | 2 |
| | DCoT | 42 | 2 |
| Phi 2 | CoT | 0 | 3 |
| | DCoT | 2024 | 2 |
| LLaMA2 7B | CoT | 0 | 2 |
| | DCoT | 0 | 3 |
| LLaMA2 13B | CoT | 42 | 3 |
| | DCoT | 42 | 3 |

tuning, so the results on the dev set are representative of the performance of our method on large-scale models.

Table 9 reports the best hyperparameters we found on the dev set. Training Phi 1.5 on DCoT takes approximately 12h, Phi 2 20h, LLaMA 7B 35h, LLaMA 13B 51h, and LLaMA 70B 30h. Training on CoT takes 9h for Phi 1.5, 15h for Phi 2, 25h for LLaMA-2 7B, 39h for LLaMA-2 13B, and 13h for LLaMA-2 70B. As expected, DCoT training is slower since the targets are longer. The parameters we use to train the models are reported in Table 8.

## C   PROMPTING

The prompts we used with GPT4o for DCoT and CoT are "Generate k different reasoning chains that answer the question. Make sure that none of the reasoning chains are repeated. Generate each reasoning chain independently, and not based on previous reasoning chains. This means that each reasoning chain must be as different from the others as possible. When generating the different reasoning chains, do so without knowledge of the answer. Each step in each of the reasoning chains must build on the previous steps in that reasoning chain. Once the required number of reasoning chains are generated, generate an answer based on the all the answers generated by all the reasoning chains." and "Generate a reasoning chain that answer the question." In both cases, after generating the CoT, we extracted the answer with the following prompt for SVAMP "Therefore, based on the solution above, extract the number that represents the answer:" and "Therefore, based on the solution

Table 10: Training datasets. The training size corresponds to our CoT generations to build the DCoT dataset.

| Dataset | Task | Train | Dev | Test | License | Source |
|---------|------|-------|-----|------|---------|--------|
| ARC | Multiple choice | 1033 | 294 | 1150 | CC BY-SA 4.0 | Link |
| BGQA | Multiple choice | 716 | 500 | 1000 | CC BY | Link |
| Coin Flip | Multiple choice | 1000 | 1333 | 3333 | mit | Link |
| CQA | Span extraction | 958 | 285 | 804 | CC BY-SA 4.0 | Link |
| GSM8K | Generation (numbers) | 1000 | 500 | 1319 | mit | Link |
| HQA | Span extraction | 1000 | 500 | 7405 | CC BY-SA 4.0 | Link |
| LLC | Generation | 350 | 50 | 100 | N/A | Link |
| Quartz | Multiple choice | 953 | 384 | 784 | CC BY-SA 4.0 | Link |
| StrQA | Boolean QA | 998 | 343 | 344 | mit | Link |

Table 11: Out of domain datasets.

| Dataset | Task | Dev | License | Source |
|---------|------|-----|---------|--------|
| AQuA | Multiple choice | 254 | Apache 2.0 | Link |
| CSQA | Multiple choice | 1220 | mit | Link |
| SVAMP | Generation (numbers) | 100 | mit | Link |
| Big Bench Hard | Multiple choice & Generation | 6511 | mit | Link |

above, select one of the options (options) as the answer to the question (just give me the option and nothing else)." for ARC and Quartz.

# D  DEV SET RESULTS

We report the mean and stardard deviation results from the validation set across threee random seeds in Table 12.

# E  MANUAL ANALYSIS

Appendix E shows two examples of how the second CoT of LLaMA 7B with DCoT corrects the first CoT.

# F  DCOT BEST $k$ PARAMETER

Table 15 shows the best $k$ (i.e., number of CoTs) per model and dataset according to the dev set.

# G  DCOT PERFORMANCE ACROSS $k$

Figure 3 shows the performance gains of `DCoT@k` against `DCoT@1`.

# H  DATA GENERATION

We report the CoT triggers used to generate the training CoTs in Table 14. To extract the answers from the CoTs, we used the following format: "{cot} Therefore, the answer (A, B, C, or D) is:" where we change (A, B, C, D) for the corresponding options of the dataset. If the dataset expects a number and not a list of options, we don't give any list of options in the prompt and extract the number with a regular expression. Lastly, for the span extraction datasets, we use the following template: "{text} {question} Answer: {answer} {cot_trigger}." The idea behind this template is to

Table 12: Dev set results using 3 random seeds. *One seed for LLaMA-2 13B Chat and 70B.

| LLM | Method | k | Avg | ARC | BGQA | CQA | GSM8K | HQA | LLC | Quartz | StrQA |
|---|---|---|---|---|---|---|---|---|---|---|---|
| Phi 1.5 | DCoT | 1 | 47.87±1.71 | 44.13±1.94 | 39.43±3.91 | 61.83±.74 | 36.07±1.70 | 38.70±3.18 | 36.00±3.46 | 71.69±1.73 | 55.13±.35 |
|  |  | 2 | 48.63±0.67 | 46.98±2.60 | 41.94±3.10 | 60.87±1.14 | 38.80±3.10 | 39.79±3.80 | 30.00±4.00 | 74.29±2.69 | 56.40±.87 |
|  |  | 3 | **48.96±0.66** | 47.32±1.66 | 42.75±1.92 | 60.75±1.15 | 39.00±1.71 | 38.19±2.81 | 32.67±7.02 | 75.42±2.38 | 55.57±1.52 |
|  |  | 4 | 48.76±0.33 | 46.78±1.14 | 43.23±2.22 | 60.16±1.32 | 38.93±3.31 | 37.33±2.92 | 32.67±7.02 | 75.60±3.32 | 55.41±1.30 |
|  | CoT |  | 47.51±1.77 | 46.60±2.38 | 36.65±1.90 | 59.55±0.61 | 37.40±3.22 | 35.28±4.22 | 36.67±9.02 | 75.07±2.36 | 52.84±2.47 |
| Phi 2 | DCoT | 1 | 63.91±2.58 | 75.21±1.84 | 45.01±3.03 | 65.39±1.57 | 56.47±1.68 | 62.44±2.63 | 62.67±16.29 | 82.88±1.09 | 57.28±2.35 |
|  |  | 2 | **65.33±2.80** | 76.46±2.52 | 46.89±3.85 | 65.69±2.12 | 57.60±1.64 | 63.71±2.18 | 66.67±9.02 | 84.10±1.36 | 56.44±3.33 |
|  |  | 3 | 65.30±1.72 | 75.87±1.42 | 48.06±1.75 | 65.90±2.02 | 58.20±1.91 | 61.66±2.06 | 68.00±5.29 | 83.91±1.18 | 56.28±3.90 |
|  |  | 4 | 64.89±2.39 | 75.91±2.72 | 49.11±2.31 | 65.92±1.01 | 57.07±1.33 | 59.86±.96 | 66.00±8.00 | 84.09±1.88 | 56.97±5.00 |
|  | CoT |  | 63.51±.71 | 74.19±.61 | 42.08±.79 | 66.92±.29 | 62.80±3.53 | 56.45±.78 | 62.71±3.00 | 77.92±7.30 | 66.74±15.54 |
| LLaMA-2 7B | DCoT | 1 | 61.28±.50 | 59.36±2.29 | 43.67±.35 | 65.31±.50 | 29.73±1.63 | 62.92±3.16 | 86.67±2.31 | 80.63±.92 | 61.96±1.45 |
|  |  | 2 | **62.46±.45** | 61.63±1.46 | 43.56±.80 | 66.05±.80 | 33.40±.80 | 63.86±1.23 | 86.67±3.06 | 82.11±1.57 | 62.38±1.21 |
|  |  | 3 | 62.37±.23 | 60.98±2.37 | 44.23±.95 | 66.65±1.21 | 33.53±.50 | 63.46±1.46 | 86.67±1.15 | 80.89±2.65 | 62.51±.86 |
|  |  | 4 | 62.42±.59 | 62.13±3.21 | 43.85±.45 | 65.98±2.72 | 33.33±.50 | 63.63±2.16 | 86.00±3.46 | 82.20±2.78 | 62.20±1.42 |
|  | CoT |  | 59.30±.54 | 56.54±3.83 | 41.91±2.32 | 59.85±3.91 | 31.93±1.42 | 57.81±3.73 | 82.67±3.06 | 79.24±2.16 | 64.42±1.52 |
| LLaMA-2 13B | DCoT | 1 | 67.30±.49 | 74.85±1.68 | 46.40±4.13 | 68.55±1.33 | 44.53±1.51 | 72.35±.93 | 81.33±3.06 | 84.89±.90 | 65.46±1.17 |
|  |  | 2 | **66.92±.59** | 73.63±1.80 | 45.74±3.50 | 67.01±1.75 | 46.93±1.22 | 72.69±.85 | 81.33±3.06 | 84.37±1.04 | 63.62±1.32 |
|  |  | 3 | 66.70±.55 | 74.95±1.50 | 45.89±3.64 | 67.26±1.47 | 45.73±.42 | 72.75±.94 | 80.67±4.16 | 83.68±1.69 | 62.71±.75 |
|  |  | 4 | 64.20±.66 | 72.41±1.21 | 43.30±3.10 | 67.12±2.19 | 39.27±2.58 | 64.20±2.43 | 79.33±1.15 | 81.68±.65 | 66.31±.68 |
|  | CoT |  | 65.41±.91 | 71.66±2.15 | 44.45±1.53 | 68.39±1.70 | 42.67±2.32 | 66.12±.82 | 82.00±5.29 | 82.37±.82 | 65.64±1.29 |
| LLaMA-2 13B Chat* | DCoT | 1 | 64.53 | 71.85 | 47.11 | 67.37 | 41.60 | 70.52 | 68.00 | 82.81 | 66.97 |
|  |  | 2 | 65.95 | 70.73 | 47.76 | 69.16 | 42.40 | 71.02 | 74.00 | 84.87 | 67.68 |
|  |  | 3 | 66.10 | 72.22 | 46.82 | 67.48 | 43.60 | 72.08 | 76.00 | 84.87 | 65.76 |
|  |  | 4 | 66.17 | 71.85 | 45.03 | 69.70 | 45.00 | 71.75 | 74.00 | 86.44 | 65.59 |
|  | CoT |  | **66.27** | 70.43 | 45.36 | **70.71** | 44.20 | 70.11 | **80.00** | 82.53 | 66.78 |
| LLaMA-2 70B* | DCoT | 1 | 66.48 | 85.80 | 36.92 | 65.89 | 56.00 | 49.78 | 78.00 | 87.00 | 72.41 |
|  |  | 2 | **68.63** | **89.04** | 38.30 | 69.57 | **66.00** | 49.78 | **82.00** | 85.99 | 68.34 |
|  |  | 3 | 68.07 | 86.64 | 38.55 | 68.87 | 60.00 | 49.78 | 80.00 | **88.00** | 72.70 |
|  |  | 4 | 68.06 | 88.94 | 32.62 | 67.21 | 66.00 | 49.78 | 82.00 | 85.00 | **72.92** |
|  | CoT |  | 66.96 | 81.69 | **44.34** | **73.59** | 56.00 | **55.94** | 76.00 | 81.99 | 66.15 |

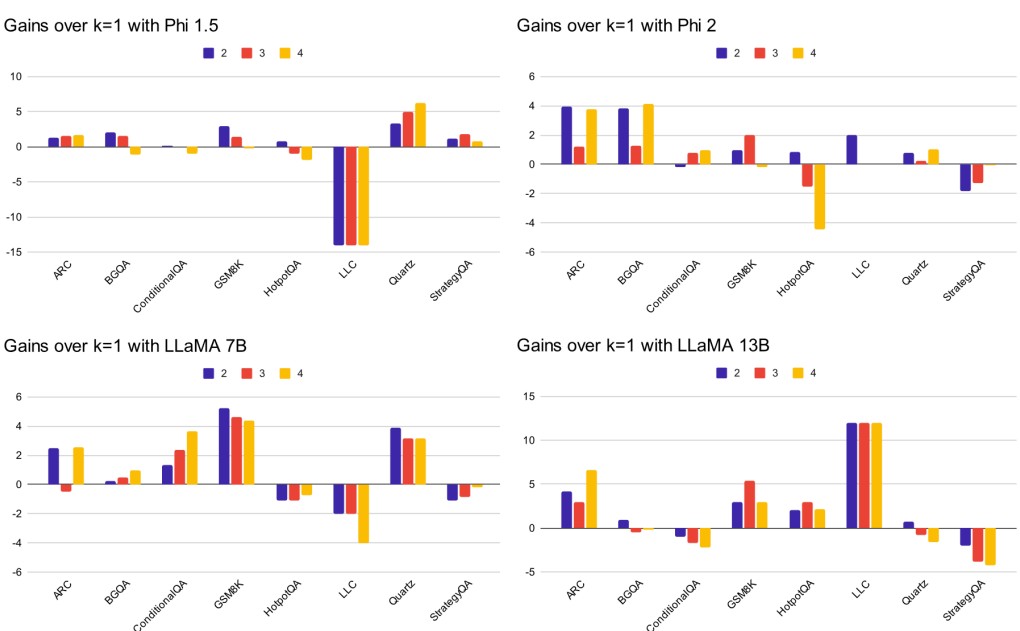

Figure 3: DCoT performance across different $k$ on the dev set.

provide the golden answer and prompt the model to generate rationales that explain that answer and use them as CoTs as in (Kim et al., 2023). The total GPT cost to generate the CoTs is $43.68.

Table 13: Examples of self-correction.

| Type | Text |
|---|---|
| 1st and 2nd CoTs are the same but the 1st CoT makes an error when generating the answer. 2nd CoT corrects that. | **Question**: Take the last letters of each words in "Billie Paloma Tanner Raul" and concatenate them.
**[Answer 1]** 1. Write down the words: Billie Paloma Tanner Raul
2. Take the last letter of the first word "Billie", which is "e".
3. Take the last letter of the second word "Paloma", which is "a".
4. Take the last letter of the third word "Tanner", which is "r".
5. Take the last letter of the fourth word "Raul", which is "l".
6. Concatenate these letters together: "ealr".
7. The result word is "ealr".
**[Answer 2]** Sure, let's break it down step by step:
1. The last letter of "Billie" is "e".
2. The last letter of "Paloma" is "a".
3. The last letter of "Tanner" is "r".
4. The last letter of "Raul" is "l".
Now, if we concatenate these letters, we get the word "earl". Therefore, the answer is "earl".
**[Final answer]** earl |
| 2nd CoT shows a different reasoing pattern and yields the correct answer. | **Question**: Cornelia likes to travel. She visited already 42 different countries. 20 of them were in Europe and 10 in South America. From the rest of the countries, only half of them were in Asia. How many Asian countries has Cornelia visited?
**[Answer 1]** 1. We know that Cornelia visited 42 countries in total.
2. We also know that 20 of those countries were in Europe and 10 were in South America.
3. Therefore, the remaining countries must be half of the total number of countries Cornelia visited, which is 42/2 = 21.
4. We know that Cornelia visited only half of the Asian countries, which means there are 21 - 10 = 11 Asian countries left to be determined.
5. Therefore, Cornelia visited 11 Asian countries in total.
**[Answer 2]** Step 1: Let's start by finding out how many countries Cornelia visited in Europe and South America. We know that she visited 20 countries in Europe and 10 countries in South America, so the total number of countries she visited in these two regions is: 20 + 10 = 30
Step 2: We know that Cornelia visited 42 countries in total. We also know that 30 of these countries were in Europe and South America. Therefore, the number of countries she visited outside of these regions is: 42 - 30 = 12
Step 3: We are told that only half of the countries Cornelia visited outside of Europe and South America were in Asia. This means that the number of countries she visited in Asia is: 12 / 2 = 6
Therefore, Cornelia visited 6 Asian countries.
**[Final answer]** 6 |

Table 14: List of CoT triggers used to generate the training CoTs. The bottom part of the table are the triggers for span extraction datasets.

| **CoT Triggers** |
| --- |
| Answer: Let's think step by step. |
| Answer: Before we dive into the answer, |
| Answer: Let's think like a detective step by step. |
| Answer: Let's think about this logically. |
| Answer: Let's solve this problem by splitting it into steps. |
| Answer: The answer is after the proof. |
| Answer: Let's differentiate using step by step reasoning . |
| Answer: Let's think step by step using inductive reasoning. |
| Answer: Let's be concise and think step by step. |
| Answer: Let's reflect on each answer option step by step. |
| Answer: Let's think step by step given every option equal consideration. |
| Answer: Let's think step by step like a scientist. |
| Answer: Let's use step by step inductive reasoning. |
| Answer: Let's work by elimination step by step. |
| Answer: Let's use step by step deductive reasoning. |
| Answer: Let's work this out in a step by step way to be sure we have the right answer. |
| because of the following reasons: |
| Justification: |
| Here's why: |
| Here is a list of the reasons: |
| Now, let's think step by step about the reasons: |

Table 15: Best number of cots ($k$ parameter) for each model and dataset in our best DCoT models according to the dev set.

| Dataset | Phi 1.5 | Phi 2 | LLaMA2 7B | LLaMA2 13B |
| --- | --- | --- | --- | --- |
| ARC | 4 | 2 | 4 | 4 |
| BGQA | 2 | 4 | 4 | 2 |
| ConditionalQA | 2 | 4 | 4 | 1 |
| GSM8K | 2 | 3 | 2 | 3 |
| HotpotQA | 2 | 2 | 1 | 3 |
| LCC | 1 | 2 | 1 | 2 |
| Quartz | 4 | 4 | 2 | 2 |
| StrategyQA | 3 | 1 | 1 | 1 |

