# OpenReview forum: "Fine-Tuning with Divergent Chains of Thought Boosts Reasoning Through Self-Correction in Language Models"
_ICLR.cc/2025/Conference — ICLR 2025 Conference Withdrawn Submission_

### Official Review · Reviewer_SbCH · 2024-10-31

**Soundness:** 2
**Presentation:** 2
**Contribution:** 2
**Rating:** 3
**Confidence:** 4

**Summary:**

This paper introduces *Divergent Chain of Thought* (DCoT), a method for inducing multiple chains of thought in a single inference call from LLMs. Specifically, the paper focuses on *DCoT fine-tuning*, where LLMs are instruction-tuned using *DCoT data*, which includes instances where a question is paired with multiple valid reasoning chains. The paper evaluates DCoT across various reasoning benchmarks, comparing its performance with LLMs instruction-tuned on standard CoT data, where each instance represents a question paired with a single valid reasoning chain. Additionally, the paper provides an analysis of the individual reasoning chains produced in a single DCoT inference, examining the soundness of subsequent chains. Finally, it discusses the implications of this analysis, arguing that DCoT promotes a self-correcting mechanism within the model.

**Strengths:**

Positive aspects include the following:
- The paper is clearly written and easy to follow.
- The paper uses open-access LLMs and reports both training parameters and hyperparameters for DCoT. Furthermore, a link to an anonymized GitHub repository is provided, thereby facilitating reproducibility.
- Experiments have been conducted across various benchmark datasets, with some models evaluated using multiple random seeds.
- The observation that DCoT@1 approximates the performance achieved by fine-tuning on CoT data, suggesting that DCoT@k could be effectively viewed as a natural extension of CoT fine-tuning.

**Weaknesses:**

The paper’s main weaknesses are:
- DCoT promotes the generation of multiple reasoning chains before arriving at a conclusion, which significantly increases token generation relative to the standard CoT approach considered. This increased computational cost is not sufficiently discussed in the paper.
- The comparison between DCoT and CoT in Tables 2, 4 and 5 seems to be based on unfair grounds. Given that DCoT involves generating multiple reasoning chains, a more appropriate, fairer comparison would be between DCoT and CoT combined with Self-Consistency (CoT + SC). While Table 2 includes this comparison, Tables 4 and 5 omit the corresponding results. The authors themselves acknowledge in lines 233-234 that the selected out-of-domain evaluation tasks benefit from self-consistency, making a DCoT vs. CoT + SC comparison much more fair. Notably, Table 2 indicates that DCoT only outperforms CoT + SC for Phi models, whereas CoT + SC generally yields better performance than DCoT with Llama-2. This is not sufficiently discussed in the paper.
- Several claims are too strong and do not seem to be sufficiently supported by the experimental evidence provided. For instance:
  - The "in-domain effectiveness of DCoT" (ll. 249-250) is overstated as "consistent and significant." (ll. 261-262). While DCoT may offer performance gains over CoT, these improvements are not “consistent” across different dataset-model pairs (e.g., Llama-2-7B on StrQA). Moreover, for several datasets, the gains are marginal, calling into question the “effectiveness” of DCoT, especially given its higher computational cost relative to CoT.
  - The statement that “DCoT is effective for any k > 1” (lines 308-309) does not align with the experimental evidence, particularly in Table 3. For instance, increasing k degrades performance for Phi 1.5, and results such as DCoT@2 outperforming DCoT@4 on Llama-2-13B indicate that performance does not always increase with k. No evidence suggests that higher values, such as k = 10, would not decrease performance compared to DCoT@1.
  - The claim that “DCoT has the ability to self-correct” (line 500) is not supported by Section 5’s analysis. For example, while Table 6 shows that generating two reasoning chains improves outcomes in 62% of cases, the remaining 38% of cases do not benefit (potentially introducing rather than correcting an error). Additionally, the experiments in Section 5.3 reveal that when a second CoT generation produces an error, a subsequent attempt does not necessarily self-correct it (A, B, B pattern). Consequently, the notion of DCoT’s self-correction ability lacks adequate empirical support. Instead, the model seems to sometimes produce a second correct CoT, while other times it produces an incorrect one.
- There exist newer, updated versions of the models evaluated in this work (e.g. Llama-3.1). Furthermore, only one model family of size > 3B and one model family of size < 3B have been considered.

**Questions:**

Suggestions for the authors:
- In several instances, the paper employs language that anthropomorphizes LLMs. For example, the expressions “a second thought can be beneficial to find the answer” (lines 199-200) and “an overthinking effect” (line 313) imply cognitive functions. While these terms may lend an intuitive appeal to describing the LLM's reasoning process, they risk creating a misleading perception of the model's actual internal workings.
- The term “significant” should be used with care. For instance, in lines 261-262, the authors state, “DCoT achieves consistent and significant performance gains compared to CoT.” However, without a statistical significance test, this terminology may be misleading and should not be used in this context.

Questions for the authors:
- How many CoTs were sampled for the self-consistency (SC) experiments, such as for CoT + SC in Table 2?
- Have the authors conducted CoT + SC experiments for models on unseen tasks (Table 4) and Big Bench Hard (Table 5)?
- How do the authors respond to the view that the model does not engage in self-correction but rather generates additional reasoning chains, which independently vary in accuracy from the preceding reasoning chain (sometimes correct, sometimes incorrect)?

---

> ### Author Response · Authors · 2024-11-25
>
> Thank you for your insightful comments. We are glad to see that the paper is easy to read, easy to reproduce, that our experiments are conducted across various benchmarks with multiple random seeds, and that our DCoT method can be viewed as a natural extension of CoT fine-tuning. Please find below our responses to the weaknesses and questions.
>
> Responses to weaknesses.
>
> W1. DCoT is more expensive than CoT
>
> Yes, DCoT is more expensive than CoT in exchange for more performance, just as CoT is more expensive than generating an answer without CoT. We believe this is a common trade-off. This is also the case with self-consistency and has been shown in prior self-consistency methods (Wang et al., 2023; Yoran et al., 2023;  Zhao et al., 2023a) and the new OpenAI o1.
>
> W2. It’s not fair to compare CoT vs. DCoT. It would be better to compare CoT + SC vs. DCoT
>
> Our goal is to compare two training regimes: CoT and DCoT. Both of them were trained with the same amount of CoTs. The only difference is how we present the CoTs to the model. In the CoT baseline, we use one CoT per data point, while in DCoT we use more than one but less data points in the end, as depicted in Figure 2. Therefore, our method is also compatible with any CoT add-ons, such as self-consistency.
>
>
> W3.1. DCoT gains are not consistent and are marginal in some cases. Is it effective then?
>
> Self-consistency can give marginal results in some cases as we show in Table 2. We believe the marginal gains of DCoT similarly only happen on minority cases, but the trend is clear, DCoT yields gains larger than 1 point in general. DCoT achieves gains for all datasets except LLC for Phi 1.5, LLC and StrategyQA for Phi 2, BGQA and StrategyQA for LLaMA2 7B, and ARC, LLC and Quartz for LLaMA2 13B. This means DCoT is beneficial for at least 75% of the tasks in all models except the 13B.
>
> W3.2. “DCoT is effective for any k > 1” is not true because increasing k can decrease performance
>
> By effective for any k > 1 we mean that DCoT@2,3, and 4 are better than DCoT@1.  However, we agree that increasing k can affect negatively because in the model’s attempt to generate a better answer, it can create a worse one. This is a well-known problem of self-reviewing methods that usually makes these methods not work  (Huang et al., 2024; Stechly et al., 2024; Tyend et al., 2023; Zhao et al., 2023a). However, Tables 2 and 3 show that in our case, we can achieve performance gains wrt k =1 with any k > 1 in most cases. Please also note Figure 12 in the appendix that shows DCoT@ k >1 for Phi 1.5 yields gains in all datasets except 2, and the performance downgrade in LLC makes the Phi 1.5’s average misleading. The following table shows the effectiveness of any k compared to k=1.
>
> | **Model** | **Effectiveness Ratio** |
> |---|---|
> | Phi 1.5 | 66.67 |
> | Phi 2 | 79.17 |
> | LLaMA 2 7B | 83.33 |
> | LLaMA 2 13B | 25 |
> | LLaMA 2 70B | 70.83 |
>
> To compute the effectiveness ratio, we calculate the number of times each model for all k > 1 is better than k =1 for every dataset.
>
> W3.3. DCoT has self-correction is not proven enough. A first correct CoT can be followed by a wrong second CoT.
>
> While this is possible, we show that the first incorrect CoT is followed by a correct CoT more often than chance. We believe this is something common in all machine learning models, where 100% accuracy is not possible. Prior works on self-correction show that this ability is not achieved by LLMs yet  (Huang et al., 2024; Stechly et al., 2024; Tyend et al., 2023). However, we show that in general, DCoT does achieve this. As we show in Table 6, even just one revision achieves gains on 62% of the cases, while the rest need more than one. Table 3 shows that in general any k > 1 is better than k =1. Section 5.3 also shows that when the model changes an answer in the second CoT, it tends to maintain it, showing that the model is not randomly generating CoTs, i.e., we did not find the pattern ABC, only AAB or ABB, which shows the model’s attempt to self-correct.
>
> W4. We should use LLaMA 3
>
> With LLaMA 3 and 3.1, we cannot do experiments across model sizes in a single model family as LLaMA 3 and 3.1 only have 8B and 70B (LLaMA 3.2 was released only one month ago). Therefore, we conducted experiments with 2 families, one for smaller models (1.3B and 2.7B) and another family for larger but still manageable sizes (7B, 13B, and core experiments on 70B). These experiments were carefully constructed and conducted over several months, before the release of LLaMA 3. However, we will work on adding these results to future works of the paper.

---

> > ### Author Response · Authors · 2024-11-25
> >
> > Responses to the questions and suggestions.
> >
> > 1. Do not anthropomorphize the model so much. “a second thought can be beneficial to find the answer” (lines 199-200) and “an overthinking effect” (line 313) imply cognitive functions.
> >
> > Thank you for your suggestion. We tried to follow the language of existing descriptions of these models (e.g., a second thought in chain of thought), which has led to  anthropomorphizing them. We will be sure to correct this.
> >
> > 2. How many CoTs were sampled for the self-consistency (SC) experiments, such as for CoT + SC in Table 2?
> >
> > Same as in DCoT, up to 4. We tune the best k for self-consistency and DCoT in the dev set, and evaluate that k on the test set.
> >
> > 3. Have the authors conducted CoT + SC experiments for models on unseen tasks (Table 4) and Big Bench Hard (Table 5)?
> >
> > As we explained above, we conduct our main experiments with greedy decoding, and we only conduct the main experiments with self-consistency to show that DCoT is compatible with any CoT addon.
> >
> > 4. "The model does not engage in self-correction but rather generates additional reasoning chains, which independently vary in accuracy from the preceding reasoning chain (sometimes correct, sometimes incorrect)"
> >
> > We partially agree with the reviewer. As we discussed in Section 6 (lines 506-509), the self-correction ability stems from the model’s attempt to generate subsequent correct CoTs. The model does not aim to identify errors in specific reasoning steps, but simply aims to generate better CoTs, and as a side-effect, it corrects prior CoTs.

---

> > > ### Comment · Reviewer_SbCH · 2024-11-26
> > >
> > > Thank you for your elaborate response to the concerns and questions raised.
> > >
> > > W1. DCoT’s increased computational cost is not sufficiently discussed in the paper.
> > >
> > > While it is true that the trade-off between computational costs and performance is a common consideration when employing techniques such as CoT or self-consistency, the paper still lacks a discussion on the additional computational costs associated with DCoT relative to its performance gains.
> > >
> > > W3.1 The "in-domain effectiveness of DCoT" (ll. 249-250) is overstated as "consistent and significant." (ll. 261-262).
> > >
> > > I am still not convinced by the author’s claim that DCoT’s "in-domain effectiveness” (ll.249-250) is both "consistent and significant." (ll. 261-262). For instance, why does DCoT perform inferior to CoT for Llama2 13B on 3 out of 8 tasks? Additionally, could the authors evaluate whether the performance improvements of DCoT over CoT are statistically significant across the full range of tasks?
> > >
> > > W3.2 The statement that “DCoT is effective for any k > 1” (lines 308-309) does not align with the experimental evidence.
> > >
> > > Thank you for providing the additional effectiveness ratios in your response. However, the assertion that "achieving performance gains for most k>1 compared to k=1" differs from the claim that "DCoT is effective for any k>1" (lines 308–309). For example, why is the effectiveness ratio for LLaMA2 13B significantly lower compared to other models? Furthermore, how can we determine the optimal k value in advance, given that the performance varies inconsistently across k values?
> > >
> > > Considering that the weaknesses highlighted have not yet been fully addressed, I will maintain my current score and recommend revising the submission.

---

### Official Review · Reviewer_eCvE · 2024-11-01

**Soundness:** 3
**Presentation:** 3
**Contribution:** 3
**Rating:** 6
**Confidence:** 4

**Summary:**

This paper introduces Divergent Chain of Thought (DCoT), a  method for enhancing the reasoning capabilities of large language models (LLMs). DCoT prompts LLMs to generate multiple reasoning chains, fostering a divergent exploration of potential solutions. Subsequently, the model analyzes these diverse chains to identify the most accurate answer through a convergent process.

Empirical validation of DCoT's efficacy is provided through extensive experimentation across various reasoning tasks, model families, and scales. Results demonstrate consistent performance improvements compared to conventional Chain-of-Thought (CoT) techniques.

This study offers valuable insights into augmenting LLM reasoning abilities by promoting internal self-evaluation and correction mechanisms, thereby contributing to the development of more reliable and robust reasoning capabilities in future language models.

**Strengths:**

The main contributions of the paper are as follows:

(a)  This method augments the traditional Chain of Thought (CoT) prompting technique by generating multiple, divergent reasoning chains before converging on a final answer.

(b)  Through rigorous experiments, the authors demonstrate that fine-tuning LLMs on DCoT datasets significantly improves their performance compared to the traditional CoT baseline across various model families and sizes (from 1.3B to 70B parameters). The improvement is consistent across a range of tasks requiring different reasoning types, including math, logic, symbolic reasoning, and multi-hop reasoning.

(c) Notably, DCoT exhibits a beneficial side effect - it enables LLMs to self-correct without external feedback or prompt optimization

**Weaknesses:**

If the authors could provide the following details it would make the paper even stronger:

(a) Diversity of CoTs Generated by Different Prompt Suffixes --  they do not directly evaluate or discuss the diversity of the CoTs generated by these different suffixes. It would be good to understand how different the CoTs are

(b) While the authors described how multiple CoTs are generated in DCoT, the provide limited information on the exact convergence mechanism used to arrive at a final answer based on these diverse reasoning chains.

(c) The results in Table 3 demonstrate that while increasing the number of generated CoTs (k) generally improves performance, there are cases where increasing k leads to a performance drop, the author should provide some explanation to that.

DCoT's performance gains are not uniform across all tasks. For instance, on StrategyQA, the only boolean question-answering dataset used, CoT sometimes outperforms DCoT. This is attributed to the limited range of possible answers in boolean tasks, where the need for divergent reasoning might be less pronounced. Similarly, the results on LLaMA-2 13B for unseen tasks are mixed, with significant gains observed only in non-math domains. These findings suggest that DCoT's effectiveness might be influenced by the nature of the task and the complexity of the reasoning required.

Increased Computational Cost Compared to CoT: Generating multiple CoTs in a single inference step inherently requires more computational resources than generating just one, as in the traditional CoT method.

Added Complexity in Training Data Generation: Creating training data for DCoT involves generating multiple correct CoTs for each question, which is a more complex and potentially time-consuming process than generating single CoTs.

**Questions:**

Please refer the previous section.

---

> ### Author Response · Authors · 2024-11-25
>
> Thank you for your valuable feedback. We are glad you find our experiments rigorous, showing significant gains compared to the baselines and that our method enables self-correction without external feedback.  Please find below our responses to the weaknesses.
>
> a. How is the diversity of the CoTs generated by different prompts in the DCoT data creation?
>
> We are not the first ones to train using multiple CoTs per question. The works of Ho et al., (2023) and Huang et al. (2023) show that even training with CoTs sampled from the same input prompt leads to performance gains compared to just using one CoT per question (note they use our CoT baseline). To maximize CoT diversity, we use the implementation of  Ott et al. (2023) to generate CoTs with different prompts. From our preliminary analysis, the CoTs are very different because they force the model to answer assuming different scenarios such as teacher-student interactions, step-by-step reasoning, analyzing each candidate answer before giving the final solution, etc. For more details, we would refer the reader to the original publication and repository of Ott et al. (2023)
>
> b. What is the convergence mechanism to arrive to the final answer?
>
> Our instruction-tuning dataset instructs the models to generate k CoTs and after the final answer (see lines 172-173). Therefore, when we prompt DCoT asking to generate k CoTs, after those k, it generates [Final answer] followed by the final answer. In this way, the model automatically converges to the final answer. We do not explicitly provide additional information to converge to the final answer as this is central to our discussion on self-correction.
>
> c. Table 3 shows that sometimes increasing k can hurt performance, why?
>
> The increased complexity of generating multiple chains of thought must be weighed against the advantage provided by the extra information in the additional chains of thought. Therefore, in the model’s attempt to generate a better answer, it can create a worse one if the overall complexity is too high. This is a well-known problem of self-correcting methods that usually makes these methods not work  (Huang et al., 2024; Stechly et al., 2024; Tyend et al., 2023; Zhao et al., 2023a). However, Tables 2 and 3 show that in our case, we can achieve performance gains wrt k =1 with any k > 1 in most cases.  Please also note Figure 12 in the appendix that shows DCoT@ k >1 for Phi 1.5 yields gains in all datasets except 2, and the performance downgrade in LLC makes the Phi 1.5’s average misleading.
>
> d. DCoT gains are not uniform/consistent across all tasks, suggesting that DCoT's effectiveness might be influenced by the nature of the task and the complexity of the reasoning required.
>
> Just as the effectiveness of CoT is not consistent across tasks and the effectiveness of self-consistency is not consistent, the effectiveness depends on the nature of the task. We observe the lowest gains on StrategyQA, a boolean QA dataset, and on Last Letter Concatenation (LCC), a symbolic reasoning task. Due to the nature of these tasks, diversity in the reasoning chains is not very useful (especially on LLC), therefore it is expected to have lower gains or even downgrades. We observe similar trends on CoT+SC, which further confirms our point.
>
> e. DCoT is more expensive than CoT
>
> Yes, DCoT is more expensive than CoT in exchange for more performance, just as CoT is more expensive than generating an answer without CoT. We believe this is a common trade-off. This is also the case with self-consistency and has been shown in prior self-consistency methods (Wang et al., 2023; Yoran et al., 2023;  Zhao et al., 2023a) and the new OpenAI o1.
>
>
> f. Creating the training data is more complex than regular CoT training.
> Our data creation is not more complex than regular CoT training and is already a common practice. The works of Ho et al., (2023) and Huang et al. (2023), which we discussed in the related works, show that fine-tuning on CoT using multiple CoTs per question is beneficial (note they use our CoT baseline).

---

> > ### Comment · Reviewer_eCvE · 2024-11-26
> >
> > Thank you for clarifying the concerns raised.
> >
> > I would like to keep my original rating.

---

### Official Review · Reviewer_nr53 · 2024-11-03

**Soundness:** 2
**Presentation:** 2
**Contribution:** 3
**Rating:** 5
**Confidence:** 4

**Summary:**

The paper investigates the impact of training LLMs with multiple chain-of-through traces.
Alternative chain-of-thought traces are generated for the training data, then filtered to only contain those that produce the correct answer.
A fine-tuned model is trained to generate multiple CoT in one sequence, given a parameter k that determines the number of CoT.
At inference time, k > 1 is given to slightly improve performance on some tasks.

**Strengths:**

The idea of the paper is potentially interesting.
The paper reports on several different base models and a number of different benchmarks, which is good to see.
There is also an attempt to investigate the underlying impact to the model, which is commendable, although there are some issues (see below).

**Weaknesses:**

It is unfortunate that only Llama2 is reported, given that Llama3 and 3.1 are available and achieve much better performance on these benchmarks.
It is unclear whether the findings generalise to the current generation of models.

In Section 3.5 it is claimed that using instruction-tuned base models would be unfair, therefore non-instruction-tuned versions are used.
It is entirely unclear why that would be unfair, if both COT and DCOT started training with an instruction-tuned base model.
In fact, the overall performance would probably be higher on the benchmarks and the results would be more realistic.
Showing improvements on an intentionally weak baseline makes the conclusions less convincing.

It seems a bit unfair to tune the k for each dataset separately, given that the COT baseline doesn't receive such tuning.

If I understand correctly, one model is trained on all the training sets at the same time.
Which is different from the normal setup of training with a particular training set, making the results uncomparable to other published work.

"The best performance of our model is achieved with more than one CoT in 25 cases out of 32 dataset × LLM combinations"
There are 4 possible values of k and at 22% of them k=1 was the best - that result isn't much better than random.

Section 5 tries to prove that the model is performing self-correction but the methodology is not convincing.
First, it is assumed that there are only two possible sources for the performance difference - self-ensembling or self-correcting - but there could be many other reasons. For example, the DCOT model could be benefitting just from having a larger number of in-domain tokens in the context at inference time. Eliminating self-ensembling does not prove that self-correcting is the reason.
Furthermore, self-ensembling also cannot be ruled out like this. Two models can still be ensembled, there doesn't need to be a majority vote, as long as there is information sharing.

In line 419 it is presented that k=2 is better than k=1 in 62% of the cases. But given that 50% would be random chance, that isn't a particularly convincing result.

Most of the performance improvements are not very large, particularly considering that they are using previous-generation base models and therefore have much room for improvement.
There really should be significance testing to actually show that the improvements are not just due to randomness.

In Section 5.2:
"Of these instances where the first reasoning chain is shared, we observe that in 45% of the cases, the second CoT of DCoT@2 exhibits a different reasoning pattern from the first."
This result isn't really showing much. If the model has 50% probability of generating a correct or incorrect CoT, then this result would be consistent with that.
These examples where specifically chosen based on the first CoT being incorrect. How does the second CoT affect cases where the first CoT is correct? If it is still 45% likely to be correct then that would imply a disadvantage instead.

**Questions:**

The prompt in Section 3.2 has an "Options" field. What are the options? This doesn't seem to be mentioned anywhere.

It is said that Llama2 70B is reported only on the dev set due to "the costs for hyperparameter tuning".
In fact, the appendix mentions that the results are not even on the full dev set but only a small subset (this should be made clear in the main paper).
Surely the training of the 70B model takes much more resources than inference? If you were able to train the model, why would you not be able to test it on the test set (without doing hyperparameter tuning, sure)?

"We then randomly sample five instances per dataset, resulting in a total of 33 samples."
33 does not divide by 5. What happened there?

---

> ### Author Response · Authors · 2024-11-25
>
> Thank you for your insightful comments. We are glad you find our ideas interesting, and our experiments extensive with multiple models and benchmarks. Please find below our response to the weaknesses.
>
> **Responses to the weaknesses.**
>
> * W1. “We should use LLaMA3. It has better performance. Would the results generalize to LLaMA3?”
>
> With LLaMA 3 and 3.1 we cannot do experiments across model sizes in a single model family as LLaMA 3 and 3.1 only has 8B and 70B (LLaMA 3.2 was only released a month ago). Therefore, we conducted experiments with 2 families, one for smaller models (1.3B and 2.7B) and another family for larger but still manageable sizes (7B, 13B, and core experiments on 70B). These experiments were carefully constructed and conducted over several months before the release of LLaMA 3. However, we will work on adding these results to future works of the paper.
>
>
> * W2. Why is it unfair to use instruction-tuned models? Overall performance should be higher and results more realistic.
>
> Instruction tuning on chain of thought has been shown to improve results, and therefore most current generation LLMs are likely to have been instruction-tuned on CoTs. Therefore, using them will be unfair.
> Our goal is to compare two instruction-tuning regimes: CoT vs. DCoT. To make the comparison fair, we fine-tuned models on the same CoTs but presented differently (i.e, CoT format vs. DCoT format; see Figure 2 for an illustration and lines 178-182 for this explanation.)
>
> * W3. Unfair to tune k for each dataset separately given that CoT doesn’t have such tuning.
>
> We show in Table 12 in Appendix that even without any tuning for k, DCoT@k with k>=2 always outperforms CoT.
>
> * W4. Each model is trained on all datasets at the same time. This makes it incomparable to other published work.
>
> We followed standard fine-tuning practices from prior works. Khashabi et al., (2020) and Tafjord & Clark (2021) fine-tune T5 on a collection of QA datasets. Similarly Wei et al., (2021), Longpre et al., (2023), and  Chung et al., (2024) fine-tune their models on multiple CoT datasets. Training on multiple datasets is required to ensure generalizability. We followed the same training practice of them and fine-tuned our models on multiple CoT datasets.
>
> UNIFIEDQA: Crossing Format Boundaries with a Single QA System (Khashabi et al., Findings 2020)
>
> General-purpose question-answering with macaw. arXiv preprint arXiv:2109.02593. (Tafjord, O., & Clark, P.; 2021).
>
> Wei, Jason, et al. "Finetuned language models are zero-shot learners." arXiv preprint arXiv:2109.01652 (2021).
>
> Longpre, Shayne, et al. "The flan collection: Designing data and methods for effective instruction tuning." International Conference on Machine Learning. PMLR, 2023.
>
> * W5. The best performance of our model is achieved with more than one CoT in 25 cases out of 32 dataset × LLM combinations. There are 4 possible values of k and at 22% of them k=1 was the best - that result isn't much better than random.
>
> As we show in Table 12 in the Appendix, on the average column, DCoT@2, 3, and 4 is always better than DCoT@1. Therefore, we believe the results are better than random.
>
>
>
> * W6. Self-ensembling vs. Self-correcting are not the only options for the performance gains from DCoT.
>
> We believe there is a misunderstanding with the concept “self-ensembling.” By this, we do not refer to an ensemble of two models, but to the effect of self-consistency (their authors call it self-ensemble of CoTs). Therefore, given multiple CoTs returned by DCoT, the gains could come from the self-ensemble of CoTs (i.e., some sort of voting mechanism) or by self-correction. Since, we observe gains from k=2, we can rule out a voting mechanism (i.e, self-consistency). Morever, since the gains from k=2 are statistically significant, we can also rule out gains from random CoT generation. Therefore, the model must be conducting some sort of implicit self-correction. The in-domain tokens at inference time further supports our hypothesis of self-correction. These tokens from the first CoT makes the generation of the second CoT aware of the first CoT, and therefore, the generation of the second CoT is not a mere random generation of a new CoT, as in self-consistency.
>
> * W7. In line 419 it is presented that k=2 is better than k=1 in 62% of the cases. But given that 50% would be random chance, that isn't a particularly convincing result.
>
> Line 419 emphasizes in that gains can be achieved with one revision, while self-consistency cannot. Therefore, our method is doing something fundamentally different from self-consistency. As we show in Table 6, just one revision achieves gains on 62% of the cases. For the remaining cases, we need more than one. Table 3 shows that, in general, any k > 1 is better than k =1.
>
> * W8. Include significance testing.
>
> Thank you for your suggestion. We will conduct them and update the manuscript.

---

> > ### Author Response · Authors · 2024-11-25
> >
> > * W9 ​​"Of these instances where the first reasoning chain is shared, we observe that in 45% of the cases, the second CoT of DCoT@2 exhibits a different reasoning pattern from the first." This result isn't really showing much. If the model has 50% probability of generating a correct or incorrect CoT, then this result would be consistent with that.
> >
> > We are sorry it seems there is a misunderstaning in this manual evaluation. We sample instances where the first CoT is wrong and the second CoT is correct, i.e., all second CoTs are correct (lines 446-447). In 45% of the cases, the second CoT exhibits a different reasoning pattern, while in the rest, it’s a similar CoT with small changes that corrects the first one.
> >
> >
> > **Responses to the questions.**
> >
> > * Q1. What is the “Options” field in the prompt?
> >
> > Some datasets we use are designed for multiple choice question answering. The options field is used to indicate the answer options given by the dataset. We apologize for not explicitly stating this in the main text.
> >
> > * Q2. Why full training of 70B but not full evaluations?
> >
> > Due to the lack of resources, we could not conduct full evaluations on 70B. Nevertheless, since our goal with the 70B is just to provide some preliminary evidence of the trend holding on 70B, we just conducted smaller evaluations.
> >
> > * Q3. "We then randomly sample five instances per dataset, resulting in a total of 33 samples." 33 does not divide by 5. What happened there?
> >
> > Thank you for noticing this. We had to remove two samples due to an error in our automatic sample process. We will mention this.

---

> > ### Comment · Reviewer_nr53 · 2024-11-26
> >
> > Thank you for your response. I will keep my original assessment.

---

### Official Review · Reviewer_WcKL · 2024-11-05

**Soundness:** 2
**Presentation:** 3
**Contribution:** 2
**Rating:** 3
**Confidence:** 4

**Summary:**

This paper introduces Divergent Chain of Thought (DCoT), which generating multiple reasoning chains in a single inference step in order to improve model reasoning. Given a reasoning problem, the approach fine-tunes models on data that includes multiple chains of thought for that problem. The approach is evaluated across different model sizes (1.3B to 70B parameters) and various reasoning tasks. The paper claims that DCoT consistently outperforms CoT baselines, works well on different model sizes, generalizes to unseen tasks, and enables models to self-correct their reasoning.

**Strengths:**

-The method is evaluated on models of varying size as well as diverse reasoning tasks.

-The approach is natural and (to the best of my knowledge) novel.

-It would be relatively straightforward to implement the method with new tasks or at a larger scale.

-The hint of self-correction in the results would be very interesting if it is real.

-The paper is well-written, and clear about experimental details.

**Weaknesses:**

I am generally concerned about the inferences drawn from the experimental results. There are a number of cases where the experimental results do not, as presented, support the authors' claims.

-The out of domain generalization results are weak. The hyperparameter k which controls the number of sampled CoTs is not estimated on a separate validation set. Performance only improves when k is post-selected for maximizing test set performance. This is not a fair way to evaluate the method. In addition, the paper does not report the performance of Llama2 70B on the out of domain tasks.

-Table 12 suggests that there is at most a very small benefit (and in most cases no statistically reliable benefit) from using DCoT@1 compared to DCoT@k for k>1. This suggests that diversity at inference-time is not reliably improving results.

-The results do not currently include 95% confidence intervals. From tables 10 and 11, many of the test sets have sample size of 1500 or less. For these datasets, the confidence intervals will be at least as wide as +/- 0.03. This means that many of the measured effects of DCoT will be non-significant.

-The DCoT@1 vs DCoT@2 comparison reported in Table 6, which is used to support the argument that DCoT models are self-correcting, does not appear to be significant. 62% of tasks (25 out of 40) show an improvement with DCoT@2, but the confidence interval includes 50%.

-The manual evaluation of self-correction only evaluates cases where DCoT@1 is incorrect while DCoT@2 is correct. It concludes from this evaluation that DCoT induces self-correction behavior. However, this ignores cases where the second reasoning chain in DCoT@2 introduces an error that was not present in the first reasoning chain. The results may not indicate true self-correction, but rather random variation in the reasoning chains. Table 7 may address this issue, but it is not clear how many of the responses have been filtered by the requirement that the first CoT in DCoT@3 matches DCoT@1.

**Questions:**

Why is self-correction only manually evaluated for Llama2 7B? It seems more likely that this ability would emerge for a larger model like Llama2 70B.

---

> ### Author Response · Authors · 2024-11-25
>
> Thank you for your insightful comments. We are glad you find our methods intuitive and novel, our evaluation very comprehensive, our paper well written with clear details of our experiments, and the hint of self-correction of our method very intersting. Please find our responses below:
>
> Responses to the weaknesses:
>
> W1.1 OOD is weak. K is not estimated on dev set.
>
> Table 4 shows the performance of DCoT for all k values, so we do not maximize performance for the test set. Using the average performance across all tasks, we see that DCoT@k > 4 is always better or similar to CoT except in Phi 1.5, which is the weakest model and probably it’s more difficult for it to generalize to OOD.
>
> W1.2. No results of LLaMA 70B in OOD
>
> Inferencing on LLaMA 70B is too costly for us, so we only provide results for the core in-domain experiments.
>
> W2. Table 12 suggest small benefit for k > 1
>
> We will conduct significance tests to show the benefits better. If we analyze the results per dataset, we can observe large gains for most datasets. The low performance on LLC makes the average results seem like small benefits. However, table 6 shows the benefits of k =2 vs. k=1 better.
>
>
> W3. and W4 The results do not currently include 95% confidence intervals. DCoT@1 vs DCoT@2 might not be significant.
>
> Thank you for the suggestion. We will conduct significance tests on all our tables.
>
>
> W4. Manual evaluation: DCoT@1 wrong vs. DCoT@2 correct might not be true self-correction but random variation in the reasoning chains.
>
> As we discussed in line 507, the self-correction ability stems from the model attempting to generate subsequent correct CoTs and not identifying errors and correcting them. We believe it’s not a simple random variation in the reasoning chains because the performance gain from k = 2 is visible across mosts tasks.
>
> Responses to the questions:
>
> Q1. Why is self-correction only manually evaluated for Llama2 7B and not 70B?
>
> Our experiments on 70B are focused on the core set of experiments and are not aimed to be comprehensive due to the computational costs required to run them. We conducted the manual evaluation on LLaMA2 7B because it’s the middle-size model and could provide us with insights that apply to the small and larger models. Manual evaluation is very time-intensive, so we are unable to conduct manual evaluation on the outputs of multiple models.

---

> > ### Comment · Reviewer_WcKL · 2024-11-26
> > **Response to authors**
> >
> > Thank you for your response.
> >
> > > Using the average performance across all tasks, we see that DCoT@k > 4 is always better or similar to CoT except in Phi 1.5
> >
> > Could you clarify what DCoT@k > 4 means?

---

> > > ### Author Response · Authors · 2024-11-26
> > >
> > > Oh, sorry, we meant DCoT@4. Thank you for reading our rebuttal carefully. Very much appreciated.

---

> > > > ### Comment · Reviewer_WcKL · 2024-11-26
> > > >
> > > > Thanks for the clarification. This argument is a type of post-selection. The choice of k=4 was not made based on validation performance / prior to observing test performance.

---

### Note · Authors · 2024-12-05

**Comment:**

We thank all reviewers for their insightful comments and productive discussions. We take note of all the feedback and will revise the paper accordingly.

**Withdrawal Confirmation:**

I have read and agree with the venue's withdrawal policy on behalf of myself and my co-authors.